# VDAC2 Mediates the Apoptosis of Cashmere Goat Hair Follicle Stem Cells Through the P53 Signaling Pathway

**DOI:** 10.3390/ani15111671

**Published:** 2025-06-05

**Authors:** Long Zhu, Yueqi Zhao, Mei Zhou, Xiaotong Guo, Yinxian Zhang, Dongjun Liu, Xudong Guo

**Affiliations:** State Key Laboratory of Reproductive Regulation & Breeding of Grassland Livestock, Inner Mongolia Univesity, Hohhot 010021, China; zhulong00012021@163.com (L.Z.); zyqzyq324@163.com (Y.Z.); 18586026786@163.com (M.Z.); 15734818467@163.com (X.G.); bizhangyinxian@163.com (Y.Z.); nmliudongjun@sina.com (D.L.)

**Keywords:** VDAC2, SHFSC, apoptosis, P53 signaling pathway

## Abstract

Apoptosis regulates tissue homeostasis by removing unwanted or damaged cells during development and aging. Apoptosis plays a key role in the cyclic cycle of the hair follicle by removing a portion of the hair follicle cells through apoptotic mechanisms to prepare the follicle for the next growth cycle. VDAC2, located in the outer mitochondrial membrane, is a key apoptosis regulator, but its role in the hair follicle of cashmere goats remains unclear. We found by proteome sequencing that VDAC2 was significantly differentially expressed in the hair follicle of the Albas cashmere goat during the growth, senescence, and resting phases. RNA-seq analysis indicated that VDAC2 affected the growth of secondary hair follicle stem cells through the P53 signaling pathway. In addition, overexpression of VDAC2 followed by a P53 inhibitor partially alleviated VDAC2-induced apoptosis. These results suggest that VDAC2 plays an important role in secondary hair follicle stem cell apoptosis.

## 1. Introduction

Cashmere is an important fibrous material produced by secondary hair follicles in cashmere goats. The number of secondary hair follicles determines the quality and yield of cashmere [1]. Hair follicles are unique features that distinguish mammals from other animals. They are micro-organs involved in important physiological activities such as body temperature regulation, body protection, and sensory perception. Throughout their life cycle, they undergo a rapid growth stage (anagen), a degeneration stage driven by apoptosis (catagen), and return to the growth stage through an intervening, relatively quiescent stage (telogen) [2]. Hair follicle stem cells (HFSCs) are present in the bulge area of the outer root sheath of hair follicles and are regarded as slow-cycling cells with the potential for multilineage differentiation and good proliferative functions [3]. HFSCs power the cyclic growth of hair follicles. Moreover, recent studies have shown that HFSCs not only receive surrounding signals, but also actively send signals to regulate the organization and function of their own niches [4,5]. The growth and development of hair follicles depends on a complex network formed through multiple pathways, such as the Wnt/β-catenin pathway [6], the Sonic hedgehog (Shh) pathway [7], the Notch pathway [8], the BMP (bone morphogenetic protein) pathway [9], and the apoptotic pathway [10].

Apoptosis is a common mechanism of cell death. Studies have shown that there are two main apoptotic pathways: the extrinsic death receptor pathway and the intrinsic mitochondrial pathway [11]. Apoptosis is controlled by the interactions between multiple molecules and is primarily responsible for eliminating unnecessary cells from the body. During the development of biological systems, cells repair and remodel tissue structures and organs by regulating cell death [12,13]. Moreover, the proliferation of HFSCs is activated through the Wnt/β-catenin and PI3K/Akt pathways, while their depletion is driven by P53/p38-induced cell death [14]. Exosomes secreted by dermal papilla cells can inhibit HFSC apoptosis and affect the growth and development of hair follicles [15,16]. Therefore, it is important to study apoptosis in HFSCs and the regulatory mechanisms underlying the inhibition of apoptosis.

VDAC2, a member of the VDAC family, is a highly conserved mitochondrial outer membrane protein that forms water channels with a diameter of 3 nm in an open state [17]. VDAC is the gateway to the mitochondria, connecting the cytoplasmic and mitochondrial compartments. It belongs to the β-barrel membrane protein family and is the most abundant protein in the outer mitochondrial membrane (MOM) [18]. VDAC can regulate molecular exchange between the cytosol and mitochondria, and its key function is to transport metabolites and other essential molecules [19]. Studies have shown that embryos with VDAC2 deficiency die during development, which is believed to be due to the role of VDAC2 in regulating apoptosis [20]. In addition, neuronal apoptosis is triggered by the mitochondrial permeability transition of pore-induced apoptosis in the neurons of adult mouse brains [21]. In our previous studies, we found that VDAC2 was significantly differentially expressed during the anagen, catagen, and telogen phases of cashmere goat hair follicle development. Based on these results, we hypothesized that VDAC2 plays an important role in HFSC growth. Currently, there are limited reports on VDAC2 expression in cashmere goat secondary HFSCs (SHFSCs), and the mechanism of action of VDAC2 remains unclear. Furthermore, it is unknown whether VDAC2 affects cashmere goat SHFSCs by regulating the P53 signaling pathway.

In this study, we investigated the effects of VDAC2 on SHFSCs. Our results suggest that VDAC2 may promote the apoptosis of SHFSCs in part through the P53 signaling pathway, based on both transcriptome changes and validation of key apoptotic regulators. These results will contribute to a better understanding of the apoptosis of cashmere goat SHFSCs, as well as the growth and development of hair follicles.

## 2. Materials and Methods

### 2.1. Animals and Ethical Statement

All animal experiments complied with the National Research Council’s Guide for the Care and Use of Laboratory Animals. All protocols were reviewed and approved by the Animal Care Committee of the Inner Mongolia University. In this study, SHFSCs were isolated from healthy 3-year-old female Arbas Cashmere goats. Cells from each goat were processed and cultured independently and used as biological replicates for all experiments (*n* = 3).

### 2.2. Cell Isolation, Culture, and Transfection

For the isolation of SHFSCs, skin tissues for cell isolation were collected from 3-year-old healthy female sheep on the YIWEI White Cashmere Goat Ranch, with hair follicles in the anagen phase. During the collection process, we attempted to minimize the pain experienced by the female sheep and disinfected the wounds in a timely manner until they recovered. Subsequently, the skin tissues were rinsed 2–3 times with phosphate-buffered saline (PBS + 2% penicillin-streptomycin), cut into 1 mm × 1 mm size and digested with a neutral protease (Colldisp-RO, Sigma, Livonia, MI, USA) for 2 h to peel off the secondary hair follicles. Internal cells were released by digestion with type IV collagenase. After digestion was terminated, centrifugation was performed, and the cell sediment was collected. The precipitate was resuspended and added to the culture medium for 2–3 weeks to obtain the SHFSCs.

The culture medium consisted of DMEM/F12 supplemented with 6% fetal bovine serum, 1% penicillin-streptomycin, 14 ng/mL epidermal growth factor (EGF) (E1257, Sigma, Livonia, MI, USA), 0.4 ng/mL hydrocortisone (H0888, Sigma, Livonia, MI, USA), and 0.5 µg/mL insulin (51500056, Gibco, Grand Island, NE, USA). The culture method was based on Yan [22].

The full-length coding sequence (CDS) of goat VDAC2 (GenBank accession XM_018042435.1) was synthesized and subcloned into the pcDNA3.1(+) expression vector. The resulting construct contained no tag. Plasmid integrity was confirmed via Sanger sequencing. The overexpression vector was transfected into SHFSC via electrofection, and the interference sequence was transfected into SHFSC using an interference kit.

SHFSCs were divided into the following four groups for transfection experiments to evaluate the effects of both upregulation and downregulation of VDAC2 and to control for non-specific effects of transfection: (1) wild-type (WT), which received no treatment; (2) VDAC2-overexpression group (OE), transfected with a pcDNA3.1 vector carrying the VDAC2 coding sequence; (3) VDAC2-interference group (Si), transfected with siRNA targeting VDAC2; and (4) negative control (NC), transfected with a non-targeting scrambled siRNA sequence.

The overexpression vectors were transfected via electroporation. The optimal electroporation conditions (250 V for 2.5 ms) were determined based on testing a range of voltages (200 V, 225 V, 250 V, and 275 V, all for 2.5 ms).

### 2.3. Total RNA Extraction, cDNA Synthesis, Primer Design, and RT-qPCR

When cell confluence in 90 dishes reached 70–80%, RNAiso (Takara, Dalian, China) was added and cells were pipetted into centrifuge tubes. Chloroform was then added, and the mixture was vortexed until the liquid became milky without phase separation. After centrifugation, the supernatant was aspirated and equal volumes of isopropanol and 75% ethanol were added. After centrifugation, the pellets were resuspended in nuclease-free water. The RNA concentration was measured and recorded, and the samples were stored at −80 °C.

cDNA was synthesized using a PrimeScript^TM^ RT reagent kit with gDNA Eraser (Takara, Dalian, China). For detection based on expression levels, TB Green^®^ Premix Ex Taq^TM^ II (Takara, Dalian, China) was used, with the goat GAPDH gene serving as the internal reference gene. Each sample was tested in triplicate. Gene primers were designed using NCBI, and all primers were commissioned to be synthesized by Hongyang Biotechnology. The detailed information is provided in Table A1 Table A1.

### 2.4. Overexpression Vector Construction and RNA Interference

To construct the VDAC2 overexpression vector we used Premix Taq (Ex Taq version 2.0 plus dye) (Takara, Dalian, China) to amplify the complete coding sequence of goat VDAC2. Using NheI (Thermo, Shanghai, China) and NotI (Thermo, Shanghai, China) as restriction enzyme sites, the VDAC2 fragment was ligated into the pcDNA3.1+ reporter vector. In addition, we performed sequencing and double-digestion verification of the recombinant plasmid to confirm the successful insertion of the fragment.

We used small interfering RNA to silence the expression of VDAC2. The siRNA for silencing VDAC2 and the siRNA negative control (NC) were synthesized by GenePharma. The specific nucleotide sequences are listed in Table A4.

### 2.5. Immunofluorescence

SHFSCs were cultured in 24-well plates for immunofluorescence analysis. When the cells reached approximately 70% confluence they were washed with PBS. Next, 4% paraformaldehyde was added to fix the cells, followed by 0.1% TritonX-100 (Sigma, Livonia, MI, USA) for permeabilization. Subsequently, 1% BSA was added for cell blocking. Finally, primary antibodies were used to incubate the cells in the dark at 4 °C overnight. The next day, cells were washed with PBST, followed by the addition of secondary antibodies and incubation at room temperature in the dark for 1 h. After washing with PBST, the cell nuclei were stained with DAPI. After another PBST wash, an appropriate amount of anti-fluorescence quenching agent was added and the slides were mounted. An inverted fluorescence microscope (AXR, Nikon, Tokyo, Japan) was used to observe and capture images of stained cells.

### 2.6. CCK-8 Assay

SHFSCs were cultured in 96-well plates for the CCK-8 assay (Beyotime, Shanghai, China). Transfection was performed when the cells reached approximately 30% confluence. A CCK-8 detection kit was used to detect the viability of SHFSCs at 24, 48, and 72 h after transfection. The absorbance at 450 nm was measured using a microplate reader.

### 2.7. EdU Assay

SHFSCs were cultured in 24-well plates for the EdU assay (C10310-2, RiboBio, Guangzhou, China). Transfection was performed when the cells reached approximately 50% confluence. After 48 h of transfection, the cells were processed according to the instructions of the Cell-Light^TM^ EDU Apollo643 In Vitor Kit. An inverted fluorescence microscope (AXR, Nikon, Tokyo, Japan) was used to observe and capture images of stained cells. Image Lab 6.1 software was used to analyze the images.

### 2.8. Cell Cycle and Cell Apoptosis Detection

For cell cycle detection (C001-50, 7 sea biotech, Shanghai, China), SHFSCs were cultured in 6-well plates and 90 dishes. Cells were thawed ahead of time, cultured for 2 days, and transfected when cells reached approximately 50% fusion. Forty-eight hours after transfection, the cells were collected using trypsin digestion. After being washed with PBS, the cells were fixed with 75% ethanol at −20 °C for 2 h. Propidium iodide containing RNase A was then added to stain the cells, which were then incubated at 37 °C in the dark for 30 min. For the treated SHFSCs, we used a flow cytometer to analyze the cells at different stages, and the data was analyzed using FlowJo 10.8.1.

For cell apoptosis detection (A005-4, 7 sea biotech, Shanghai, China), SHFSCs were cultured in 6-well plates and 90 dishes for cell cycle detection. Transfection was performed when the cells reached approximately 50% confluence. Forty-eight hours after transfection, the cells were collected using trypsin digestion. After washing with PBS, the cells were resuspended in 1× Binding Buffer. Annexin V-FITC was then added to stain early apoptotic cells in the dark. After incubation at room temperature for 15 min, propidium iodide (1:1000) (D8417, Sigma, USA) was added to stain late apoptotic cells. For the treated SHFSCs, we used a flow cytometer to analyze the apoptotic cells at different stages and analyzed the data using FlowJo 10.8.1.

### 2.9. Total Protein Extraction and Western Blotting

SHFSCs were cultured in 6-well plates and 90 dishes for protein sample collection. Transfection was performed when the cells reached approximately 50% confluence. Forty-eight hours after transfection, the cells were collected using trypsin digestion. An appropriate amount of mammalian protein extraction reagent and protease inhibitors were added to lyse the cells and collect the proteins (CWBIO, Beijing, China). A BCA protein quantification analysis kit was used to determine protein concentration (Thermo Fisher Scientific, Waltham, WA, USA). After determining the concentration, 5 × SDS lane marker loading buffer was added, and the proteins were denatured by boiling for 10 min. After the proteins were denatured, a 12% polyacrylamide gel was used for electrophoresis to obtain the target proteins. The voltage of the samples in the stacking gel was 90 V, which changed to 130 V when the samples entered the separating gel. A protein transfer apparatus was used to transfer the target proteins onto the NC membranes (protein transfer conditions were 0.2 A, 0.5 KD/min). The target proteins were transferred onto nitrocellulose membranes and blocked with 5% skim milk powder for 1 h. The NC membranes were washed with TBST and then incubated with primary antibodies overnight at 4 °C. The following day, the primary antibodies were recovered and the NC membranes were washed with TBST and incubated with secondary antibodies for 1 h (Abcam, Cambridge, UK, ab6802, 1:10,000). Pierce ECL Western blotting substrate (Thermo Fisher Scientific) was dropped onto NC membranes and exposed to a fully automatic chemiluminescence imaging system to obtain images of the target proteins. The gray values of the target protein bands were detected using ImageJ v1.8.0 software to measure their expression levels.

### 2.10. Transcriptome Sequencing Analysis

SHFSCs were isolated for transcriptome sequencing. We used 6-well plates and 90 dishes to culture the SHFSCs and collect the samples required for transcriptome sequencing. The overexpression and wild-type groups were cultured in 90 dishes, and the interference group was cultured in 6-well plates. Cell samples were collected after 48 h. Transfection was performed three times, and three biological replicates were obtained for each group for a total of nine samples. After collecting the cell samples, we commissioned Shanghai OE Biotech Co., Ltd. (Shanghai, China) for transcriptome sequencing.

### 2.11. Inhibition of P53

As it was previously found that VDAC2 could affect the apoptosis of SHFSCs through the P53 signaling pathway, we attempted to identify a drug to determine whether it could inhibit the occurrence of cell apoptosis. Pifithrin-μ (PFT), an inhibitor of P53, stabilizes the mitochondrial membrane and inhibits the release of cytochrome C, thereby impeding the intrinsic apoptotic pathway. When its working concentration is 10 μM, it can inhibit the binding of P53 to mitochondria by reducing the affinity of P53 for the anti-apoptotic proteins Bcl-xL and Bcl2 [23]. We added PFT to wild-type SHFSCs and used RT-qPCR to detect the expression levels of P53.

### 2.12. Statistical Analysis

We used GraphPad Prism 9.5 Software for statistical analysis. Data were considered statistically significant at *p* < 0.05, (*), *p* < 0.01 (**), or *p* < 0.001 (***). All experiments were performed with three biological replicates (*n* = 3). The Western blot data shown represent one of three independent experiments. Quantitative analysis of band intensities was performed using ImageJ v1.8.0, and the mean ± SEM from three biological replicates was calculated.

### 2.13. Visualization and Software

Visualization of transcriptome data, including volcano plots, heat maps, and GO/KEGG enrichment plots, was performed using R software (version 4.2.0) with the ggplot2, pheatmap, and clusterProfiler packages. Flow cytometry data were analyzed and visualized using FlowJo software (v10.8.1). Densitometry analysis of Western blot bands was conducted using ImageJ v1.8.0. All bar graphs and statistical charts were generated with GraphPad Prism software.

## 3. Results

### 3.1. Isolation, Purification, and Identification of SHFSCs in Cashmere Goats

HFSCs are pluripotent stem cells located in the bulge area. The descendant cells migrate upward to the epidermis and downward to form a hair shaft. Moreover, HFSCs are one of the widely used model systems for studying adult stem cells [24,25,26]. Secondary hair follicles were isolated from the tissues through trypsin digestion (Figure 1A), and hair bulbs were obtained. After adherent culture, the cells gradually migrated out of the hair bulbs (Figure 1C). Because the isolated cells included SHFSCs and secondary dermal papilla cells, we used trypsin for multiple short-term digestions to obtain secondary HFSCs with relatively high purity (Figure 1B). We then identified the isolated cells using immunofluorescence. The results showed that the surface markers K14, K15, LGR5, and ITGβ1 of SHFSCs were immunofluorescence-positive (Figure 1D). K14 and K15 are keratin family markers of epithelial stem cells [27,28], LGR5 is a classical stemness marker in the hair follicle bulge [29], and ITGβ1 plays a key role in HFSC adhesion and maintenance [30]. Taken together, these results indicated that SHFSCs were isolated with relatively high purity.

### 3.2. Impact of VDAC2 on the Growth of SHFSCs

To investigate the function of VDAC2, we constructed a VDAC2 overexpression vector and transfected it into SHFSCs via electroporation (Figure A1). The expression of VDAC2 mRNA and protein in SHFSCs was significantly increased following transfection (Figure 2A). Additionally, we designed and synthesized three small RNA interference sequences and examined their effects on the expression of VDAC2 in SHFSCs (Table A4). The RT-qPCR results indicated that the siR-568 sequence at a concentration of 100 nM was the most effective in reducing the expression of VDAC2 (Figure 2B).

Then, we used CCK-8, EdU, Ki67, immunofluorescence, and flow cytometry to detect the impact of VDAC2 on SHFSCs. Ki67 is a nuclear protein that serves as a well-established marker of cellular proliferation. It is expressed during active phases of the cell cycle but is absent in quiescent cells [31]. The results of CCK-8, EdU, Ki67, and immunofluorescence assays showed that VDAC2 overexpression inhibited cell viability and proliferation, while VDAC2 knockdown promoted these processes (Figure 2C–E). Flow cytometry analysis (Figure 2F) revealed that VDAC2 knockdown led to a significant increase in the S-phase cell population and a corresponding decrease in the G0/G1 phase, suggesting enhanced cell cycle progression. In contrast, no statistically significant difference in cell cycle distribution was observed between the VDAC2-overexpression group and its corresponding control. These results indicate that VDAC2 knockdown may promote cell cycle entry, whereas its overexpression does not significantly alter cell cycle progression under the current conditions. When considered together with the CCK-8 and EdU results, VDAC2 appears to negatively regulate SHFSC proliferation (Figure 2F,G). Overall, the findings suggest that VDAC2 plays an important role in regulating apoptosis in SHFSCs.

### 3.3. Impact of VDAC2 on Downstream Genes

To further investigate the molecular mechanism of VDAC2 in SHFSC apoptosis, we performed transcriptome sequencing on SHFSCs with VDAC2 overexpression and knockdown, using wild-type cells as controls. The results of the transcriptome sequencing are shown in Table A2 and Table A3. Compared to the goat reference genome, the alignment efficiency of the total reads of the nine samples exceeded 97%. Meanwhile, the correlation analysis showed a high similarity among the replicate samples within the same group (Figure 3). These results indicate that the data quality was high and could be used for further analysis to identify the differentially expressed genes (DEGs) in the two groups (VDAC2-OE vs. WT and VDAC2-Si vs. WT). Based on the differential screening conditions of a q-value < 0.05 and a fold change greater than 2, a total of 630 DEGs were obtained in the VDAC2-OE group, among which 404 were upregulated and 226 were downregulated. In the VDAC2-Si group, 401 DEGs were identified, of which 106 were upregulated and 295 were downregulated (Figure 3A).

Volcano plots, Venn diagrams, and clustered heat maps were constructed to directly observe the differences in gene expression levels and their statistical significance (Figure 3B,D,E). The results showed that the DEGs were clearly distributed, further demonstrating the reliability of the sequencing data. In addition, we randomly selected five upregulated genes and five downregulated genes from both the VDAC2-OE vs. WT and VDAC2-Si vs. WT groups for verification of differential expression (Figure 3C). The RT-qPCR results showed that the expression trends of the 20 genes were consistent with the sequencing results. These results indicated that the transcriptome sequencing data could be used in the subsequent step of the analysis. GO analysis showed that these DEGs were related to biological processes, cellular components, and molecular functions (Figure 3F).

### 3.4. Impact of VDAC2 on Signaling Pathways

To explore the impact of VDAC2 on the P53 signaling pathway, we performed KEGG enrichment analysis on DEGs from transcriptome sequencing and identified four key apoptosis-related genes within the pathway (Figure 4A). We detected the expression levels of key genes (P53, BAX, BCL2, and Casp3) in the P53 signaling pathway in SHFSCs at the mRNA and protein levels after overexpression and knockdown of VDAC2 (Figure 4B,C). The results of RT-qPCR and Western blotting showed that Casp3 expression was enhanced by VDAC2 overexpression. Notably, Casp3 is usually used as a biomarker for detecting cell apoptosis [32]. Although apoptosis-related GO terms were not among the most enriched entries in GO analysis, several DEGs were found to be associated with cell death and survival processes. More importantly, KEGG enrichment analysis highlighted the P53 signaling pathway, which includes key pro-apoptotic regulators such as P53, Bax, Casp3, and anti-apoptotic gene Bcl2.

These findings, supported by follow-up experimental validation, led us to further explore the involvement of apoptosis in the effect of VDAC2 on SHFSCs. Taken together, these results indicated that VDAC2 regulates SHFSC apoptosis through the P53 signaling pathway.

### 3.5. Rescue of SHFSC Apoptosis Induced by VDAC2 Overexpression Using PFT

As it was previously found that VDAC2 could affect the apoptosis of SHFSCs through the P53 signaling pathway, we attempted to identify a drug to determine whether it could inhibit the occurrence of cell apoptosis. PFT, an inhibitor of P53, stabilizes the mitochondrial membrane and inhibits the release of cytochrome C, thereby impeding the intrinsic apoptotic pathway. When its working concentration is 10 μM, it can inhibit the binding of P53 to mitochondria by reducing the affinity of P53 for the anti-apoptotic proteins Bcl-xL and Bcl2 [23]. We added PFT to wild-type SHFSCs, used RT-qPCR to detect the expression levels of P53, and determined that the optimal addition time was 24 h (Figure 5A). After adding PFT to SHFSCs transfected with the VDAC2 overexpression vector, RT-qPCR and Western blotting results showed that the expression levels of P53 and Casp3 were both decreased at the mRNA and protein levels (Figure 5B–D). These results further prove that VDAC2 affects the apoptosis of SHFSCs through the P53 signaling pathway.

## 4. Discussion

In mammals, HFSCs periodically activate hair follicles, enabling the cyclic regeneration of hair follicles and thus the growth of hair [33,34,35]. Therefore, studying the regulation of HFSC growth will help us to understand the growth and development of hair follicles. In our previous studies, we found that VDAC2 was significantly differentially expressed during the anagen, catagen, and telogen phases of cashmere goat hair follicle development. These results prompted us to investigate whether VDAC2 plays a role in HFSC apoptosis. In previous studies, it was found that VDAC1- or VDAC3-knockout mice did not exhibit obvious or only mild phenotypes [36,37,38]. Embryos with VDAC2 deficiency die during development, and this embryonic lethality is widely attributed to the role of VDAC2 in regulating apoptosis [20]. Chin et al. determined from a whole-genome CRISPR/Cas9 screen that VDAC2, rather than Bak, is crucial for Bax function. Deleting VDAC2 disrupts the association of Bax and Bak with the mitochondrial complex containing VDAC1, VDAC2, and VDAC3, but selectively inhibits Bax-regulated apoptosis [39]. VDAC2 knockout reduces Bax expression but has no effect on BAK [40,41,42]. In melanoma cells, the downregulation of Nedd4 by shRNA rescues the elimination of VDAC2/3 proteins induced by erastin and increases the sensitivity of melanoma cells to erastin, thereby regulating ferroptosis in melanoma cells [43]. Therefore, we speculated that VDAC2 might regulate SHFSC apoptosis through Bax. In this study, we examined the impact of VDAC2 on the apoptosis of SHFSCs and found that the overexpression of VDAC2 promoted the apoptosis of SHFSCs, whereas the knockdown of VDAC2 had the opposite effect. These results indicate that VDAC2 promotes the apoptosis of SHFSCs in vitro.

Through the analysis of DEGs via RNA-seq, we preliminarily elucidated the potential functions of VDAC2 in the growth process of SHFSCs. Based on GO enrichment analysis, we found that some DEGs were related to apoptotic processes, such as P53 [44], Bax [45], Bcl2 [46], and Casp3 [47], which have been reported to regulate the apoptotic process. Therefore, we speculate that VDAC2 affects the apoptosis of SHFSCs by influencing the expression of these genes. In addition, KEGG enrichment analysis showed that DEGs were clustered in the P53 [48] and TNF signaling pathways [49], which play important roles in the apoptotic process. These results led us to propose that VDAC2 may affect apoptosis through these signaling pathways.

The pathway enrichment analysis revealed that some genes were related to the P53 pathway. This finding drew our attention to the relationship between VDAC2 and P53 pathways in SHFSCs. Apoptosis usually occurs during development and aging and serves as a mechanism for maintaining the homeostasis of cell populations in tissues. Apoptosis also acts as a defense mechanism [50]. The tumor suppressor activity of P53 is largely attributed to its ability to induce cell death (including apoptosis) through both transcription-dependent and transcription-independent mechanisms. Notably, P53 can regulate the expression of multiple genes involved in multistep mitochondrial apoptosis, including the activation of pro-apoptotic BCL-2 proteins and the inhibition of anti-apoptotic BCL-2 proteins, MOMP (mitochondrial outer membrane permeabilization), cytochrome C release, apoptosome assembly, and caspase activation [51]. Therefore, we examined the effect of VDAC2 on the P53 pathway.

Multiple components of the P53 signaling pathway, including Bax and Bcl2, were regulated by VDAC2. Caspase-3, which is usually used as a biomarker for detecting apoptosis [32], was also regulated. Therefore, the results demonstrated that VDAC2 regulates SHFSC apoptosis likely through the activation of the P53 pathway. Although both the P53 and TNF signaling pathways were enriched in our KEGG analysis, we chose to focus on the P53 pathway for further investigation for several reasons: First, several core pro-apoptotic genes in the P53 pathway were strongly differentially expressed and validated at both transcript and protein levels. Second, the mitochondrial localization of VDAC2 suggests a closer functional link with P53-mediated intrinsic apoptosis, rather than the TNF-driven extrinsic pathway. Third, previous studies have shown that VDAC2 interacts with BAX, a key downstream effector of P53, which further supports this mechanistic direction. Nonetheless, we recognize that TNF signaling also plays an important role in cell death regulation, and future work will aim to explore its potential involvement in VDAC2-mediated apoptosis.

Although our current study focused on the P53 signaling pathway, other apoptosis-related pathways, including TNF signaling and Wnt/β-catenin, were among the KEGG-enriched terms (Figure 4A). Given the central role of Wnt/β-catenin signaling in hair follicle stem cell proliferation and the involvement of TNF signaling in inflammatory apoptosis, VDAC2 may modulate SHFSC fate through additional mechanisms beyond P53. Further studies are needed to elucidate these alternative pathways and their potential crosstalk with P53. The pro-apoptotic role of VDAC2 observed in our study is consistent with findings in other cell types. For example, Chin et al. [39] demonstrated that VDAC2 is essential for BAX-mediated apoptosis in mouse embryonic fibroblasts and human tumor cells. Using a genome-wide CRISPR/Cas9 screen, they found that deletion of VDAC2 disrupted the mitochondrial localization and apoptotic function of BAX, but not BAK. Moreover, VDAC2 deficiency impaired chemotherapy-induced apoptosis and promoted tumor development in vivo. These findings highlight a unique and non-redundant role of VDAC2 in facilitating BAX activation, further supporting our conclusion that VDAC2 promotes SHFSC apoptosis through the intrinsic mitochondrial pathway.

Since VDAC2 can regulate the apoptosis of SHFSCs by activating the P53 signaling pathway, the question is whether this process can be inhibited. PFT, as an inhibitor of P53, can stabilize the mitochondrial membrane and reduce the release of cytochrome C, thereby inhibiting the intrinsic pathway in apoptosis. When PFT was applied in the context of VDAC2 overexpression, both Bax and Casp3 were inhibited. Thus, we speculated that PFT could, to some extent, rescue the apoptosis induced by VDAC2 overexpression.

In summary, our study showed that VDAC2 overexpression promotes apoptosis after activating the P53 pathway in SHFSCs. In addition, PFT can, to some extent, rescue the apoptosis caused by the overexpression of VDAC2. These findings suggest that VDAC2 promotes apoptosis in SHFSCs by activating the P53 signaling pathway.

We found that VDAC2 promotes SHFSC apoptosis via the P53 signaling pathway through transcriptome analysis and verified the expression changes in key genes, including Bax, P53, Casp3, and Bcl2, using qPCR and Western blotting. However, this study had some limitations. First, the sample size of this study was relatively small, which may have introduced some statistical bias. Second, although we verified several key genes in the P53 pathway, the effect of VDAC2 on SHFSC apoptosis may involve other unexplored signaling pathways. We found that VDAC2 promotes SHFSC apoptosis via the P53 signaling pathway through transcriptome analysis and verified the expression changes in key genes, including Bax, P53, Casp3, and Bcl2, using qPCR and Western blotting. Although we verified several key genes in the P53 pathway, the effect of VDAC2 on SHFSC apoptosis may involve other unexplored signaling pathways. More extensive studies are needed to reveal the comprehensive mechanism.

## 5. Conclusions

We preliminarily examined the effect of VDAC2 on SHFSCs and found that VDAC2 promoted apoptosis in SHFSCs. By combining transcriptome sequencing and experimental verification, we revealed that VDAC2 affects SHFSC apoptosis by influencing the P53 signaling pathway. Our study determined the impact of VDAC2 on SHFSCs, providing a further understanding of the involvement of VDAC2 in cashmere growth.

## Figures and Tables

**Figure 1 animals-15-01671-f001:**
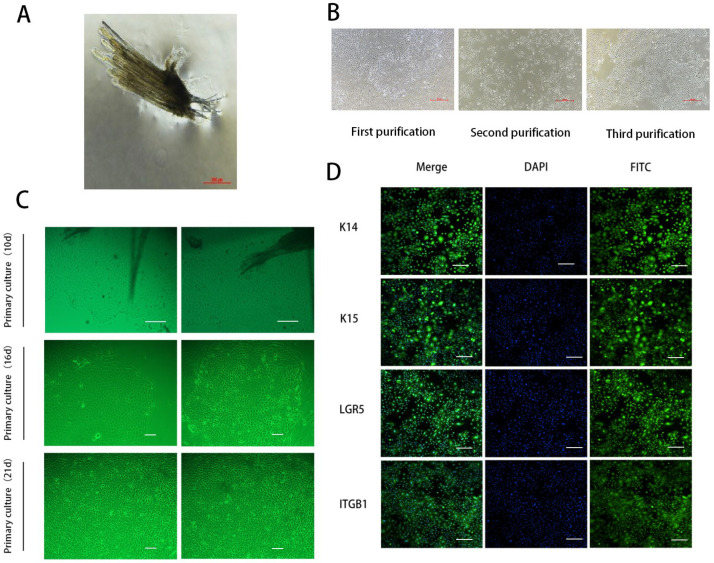
Isolation, purification, and identification of secondary hair follicle stem cells (SHFSCs). (**A**) Secondary hair follicles of cashmere goats. Scale bar = 500 µm. (**B**) SHFSCs were purified three times. Scale bar = 500 µm. (**C**) SHFSCs in primary culture for 10, 16, and 21 d. Scale bar = 100 µm (**upper**) and 500 µm (**lower**). (**D**) Expressions of K14, K15, LGR5, and ITGβ1 in SHFSCs. Scale bar = 100 µm.

**Figure 2 animals-15-01671-f002:**
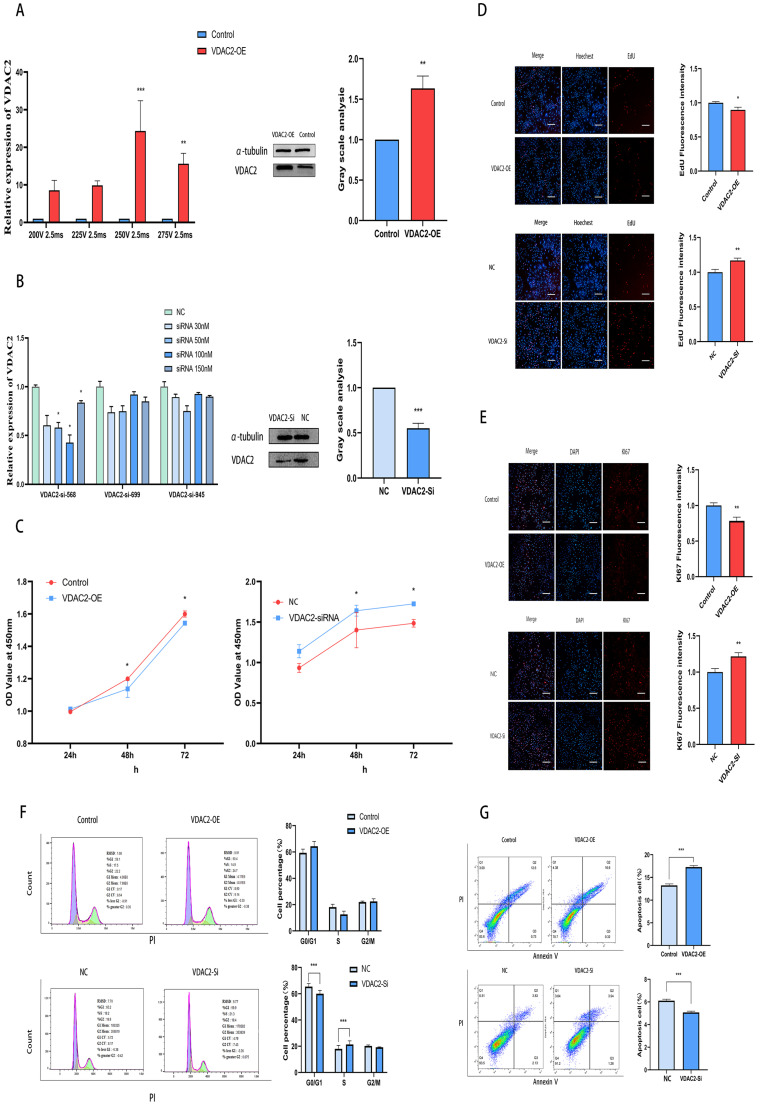
VDAC2 promotes the apoptosis of SHFSCs. (**A**) Detection of the optimal transfection conditions for the VDAC2 overexpression vector in SHFSCs and the mRNA and protein levels after VDAC2 overexpression. (**B**) Detection of the VDAC2 mRNA and protein levels after VDAC2 interference in SHFSCs. (**C**) CCK-8 assay after overexpression and interference of VDAC2 in SHFSCs. (**D**) EdU assay after overexpression and interference of VDAC2 in SHFSCs. Scale bar = 500 µm. (**E**) Ki67 immunofluorescence assay after overexpression and interference of VDAC2 in SHFSCs. Scale bar = 500 µm. (**F**) Cell cycle detection after overexpression and interference of VDAC2 in SHFSCs. (**G**) Cell apoptosis detection after overexpression and interference of VDAC2 in SHFSCs. (* *p* < 0.05; ** *p* < 0.01; *** *p* < 0.001).

**Figure 3 animals-15-01671-f003:**
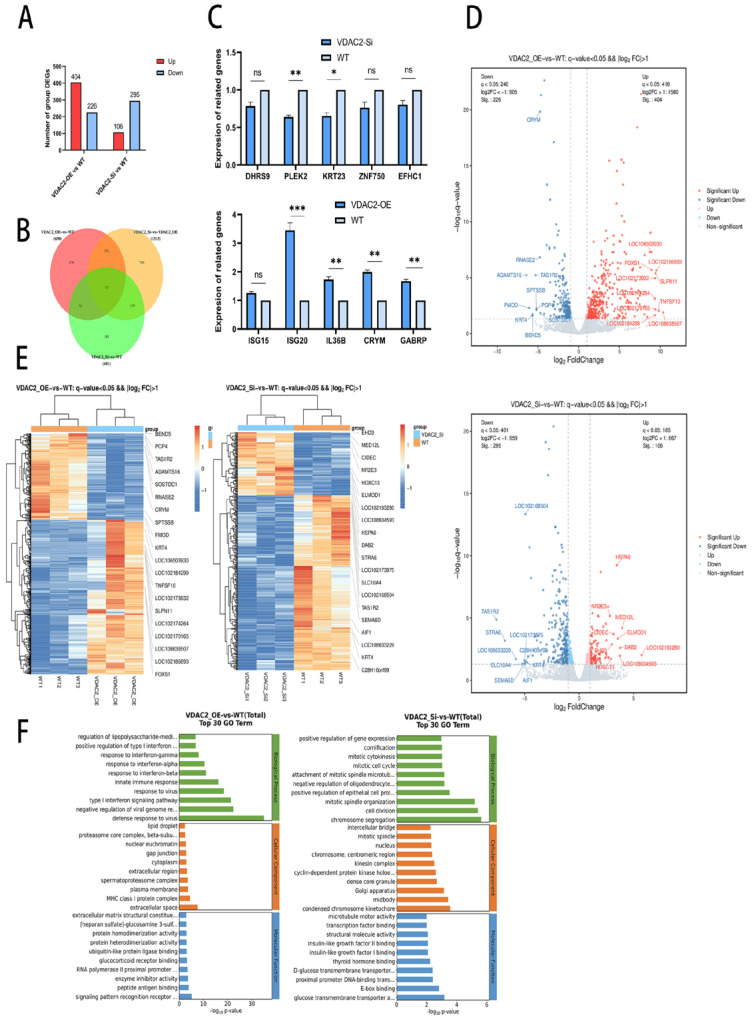
Analysis of differentially expressed genes (DEGs). (**A**) Statistical chart of the number of DEGs among different groups. (**B**) Venn diagram of the common DEGs among groups. The number of genes in each region is marked at its corresponding position. (**C**) Verification of DEGs via QRT-PCR. (**D**) Clustered heatmap of DEGs. (**E**) Volcano plot analysis of DEGs. Red, upregulated genes; blue, downregulated genes. (**F**) GO analysis of the DEGs in the VDAC2-OE, VDAC2-Si, and WT groups. (ns: *p* > 0.05; * *p* < 0.05; ** *p* < 0.01; *** *p* < 0.001).

**Figure 4 animals-15-01671-f004:**
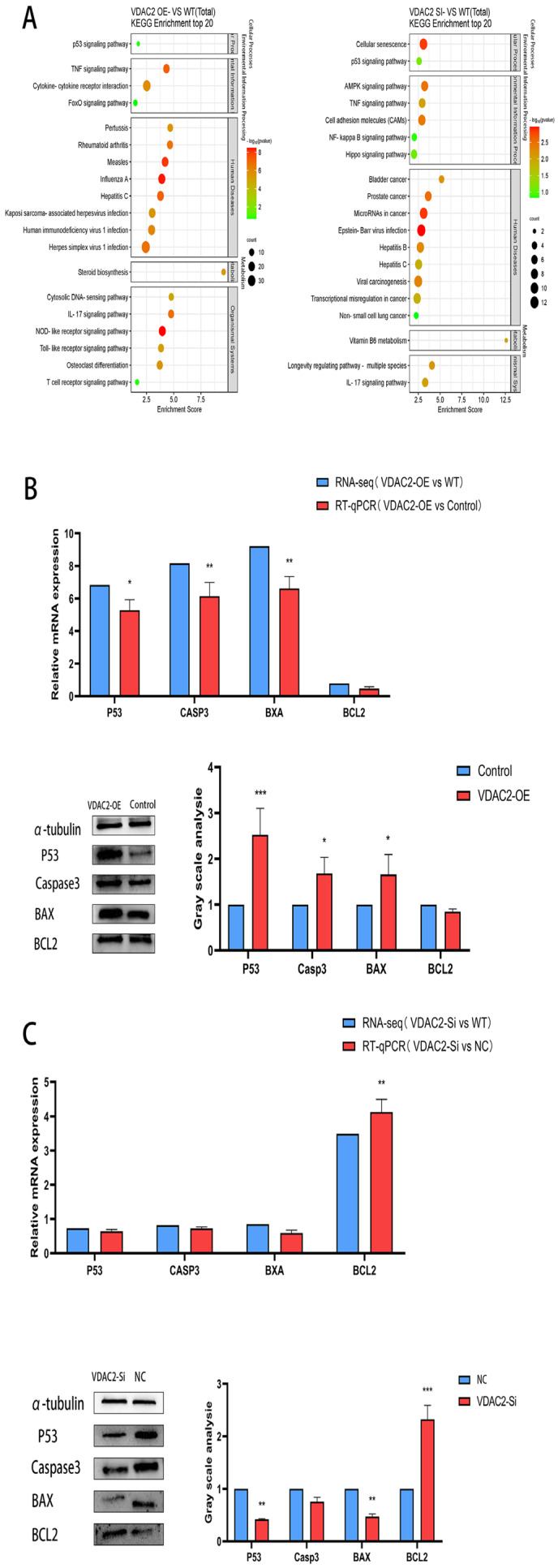
VDAC2 regulates SHFSC apoptosis through the P53 signaling pathway. (**A**) KEGG enrichment analysis of the DEGs obtained after VDAC2 overexpression and interference. (**B**) mRNA and protein expression levels of the key genes in the P53 signaling pathway after VDAC2 overexpression in SHFSCs. (**C**) mRNA and protein expression levels of the key genes in the P53 signaling pathway after VDAC2 interference in SHFSCs. (* *p* < 0.05; ** *p* < 0.01; *** *p* < 0.001).

**Figure 5 animals-15-01671-f005:**
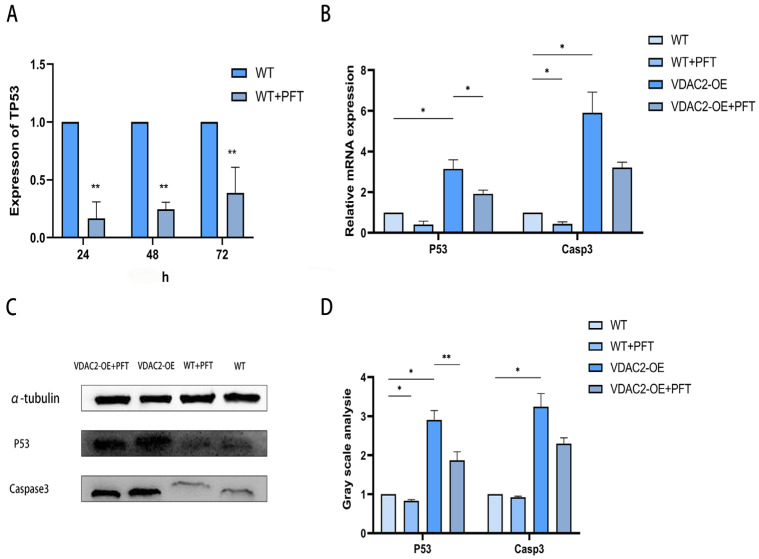
Rescue of SHFSC apoptosis induced by VDAC2 overexpression using PFT. (**A**) Detection of the mRNA expression level of P53 after adding PFT. (**B**) Detection at the mRNA level after transfection in SHFSCs. (**C**,**D**) Detection at the protein level after transfection in SHFSCs. (* *p* < 0.05; ** *p* < 0.01).

## Data Availability

The datasets used and analyzed in this study are available from the corresponding author upon reasonable request.

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
