# Peer review of "VDAC2 Mediates the Apoptosis of Cashmere Goat Hair Follicle Stem Cells Through the P53 Signaling Pathway"

_animals, 2025, doi:10.3390/ani15111671_

Round 1

Reviewer 1 Report

Comments and Suggestions for Authors

The study titled VDAC2 mediates the apoptosis of cashmere goat hair follicle stem cells through the P53 signaling pathway explored the role of VDAC2 in cashmere goat secondary hair follicle stem cells (SHFSCs). The authors claimed that VDAC2 affects the growth of secondary hair follicle stem cells through the p53 signaling pathway. And they found that the addition of pifithrin-μ (PFT), a p53 inhibitor, partially rescued SHFSCs apoptosis.

The research design is logical, and the experimental data are generally robust, but there are areas for improvement. Below is my assessment:

  1. Line 101: How many times you rinsed the skin tissues in PBS? Are there antibiotics (e.g., Pen-Strep) added in PBS?
  2. Line 102: What do you mean by “a neutral protease”? Please specify the exact name, CatLog number and the producer of this protease.
  3. Lines 99-104: How big are the skin tissues? Did you further cut them before type IV collagenase treatment? Usually, the smaller the skin tissues, the better for enzyme digestion. How long was the digestion of the skin tissues using type IV collagenase?
  4. Line 110: “references” should be “reference”.
  5. Line 122: Please delete the space between “PrimeScript” and “™ RT”.
  6. Line 137: The authors should specify which table it is.
  7. Lines 138-148: Please provide the details of your IF protocol, e.g., how long you have exactly blocked the samples using 1% BSA; the exact treatment duration of the 1st and 2nd antibodies? How many times you have wased the sample using PBST before the 2nd body treatment? Please specify the exact model of Nikon microscope that you used for imaging.
  8. Lines 154-160: Please specify the exact model of Nikon microscope that you used for imaging.
  9. Lines 161-178: How many hours it takes the SHFSCs to grow 50% confluency? And what is the confluency after 48h of transfection? Why the authors choose 48h but not 24 and 36h?
  10. Line 175: Please specify the exact dilution factor of propidium iodide for staining.
  11. Lines 201-208: Why the OE and WT groups were culture in 90 dishes, while the interference group was cultured in 6-well plates? What is the difference between 90 dishes and 6-well plate?
  12. Line 223: Clarify how "relatively high purity" was quantified (e.g., % positive for markers).
  13. Lines 224-226: Please cite relative references of these markers (K14, K15, LGR5, and ITGβ1) of SHFSCs. Provide statistical data for the ratio of the relative markers (e.g., K14) out of the total cells (DAPI). And specify how many times you have replicate this experiment. Compare the K14 positive ratio before and after the purification of SHFSCs.
  14. Figure 1C: Please provide high resolution bright field images. The authors should specify which image correlate with “upper” and “lower”, for example, they can use “Primary culture (10d)”, “Primary culture (16d)” and “Primary culture (21d)” in the legend instead of “upper” and “lower”.
  15. Figure 2A only shows that the 250V 2.5 ms yield the highest expression level of VDAC2, did they compare the protein levels between these conditions?
  16. Please discuss the details of your flow cytometry results in line 244 and Figure 2F.
  17. The authors should explain the role of Ki67 and why they chose this gene as the marker of apoptosis. Relevant references should be cited some to support your results.
  18. Delete Chinese characters in Figures 2C and 5A.
  19. Figure 2F: There is no significant difference between the Control and the VDAC2-OE group, the authors cannot conclude that “VDAC2 affects the apoptosis of SHFSCs by regulating the progression of the G0/G1 and S phases of the cell cycle” in line3 245-246.
  20. Are there any pathways related to apoptosis in your GO analysis (Figure 3F)? If not, please explain what makes you turn to apoptosis based on this RNA-seq data?
  21. The enrichment score and the p-value of p53 signaling pathway is very low in Figure 4A, the statement of “Our results demonstrate that VDAC2 promotes the apoptosis of SHFSCs by activating the p53 signaling pathway” (lines 85-86) and “RNA-seq analysis revealed that VDAC2 influenced SHFSC growth through the p53 signaling pathway” (lines 31-32) it is not convincing.
  22. Line 311-315: The authors should explain what is PFT in detail and cite relevant references?
  23. Have you done any concentration gradient experiments before adding PFT to your cells?What concentrations of PFT did the authors use in this study?
  24. The authors should specify the number of animals used and replicates.
  25. The number of biological replicates (n=?) is not clearly stated, and some data (e.g., Western blots) appear to show representative results only.
  26. The discussion does not fully address alternative explanations. The authors should also discuss whether VDAC2 might influence other apoptosis-related pathways (e.g., TNF, Wnt/β-catenin).
  27. While the study shows that VDAC2 promotes apoptosis via the p53 pathway, it does not elucidate how VDAC2 activates p53 (e.g., effects on mitochondrial membrane potential, cytochrome C release). The authors can measure mitochondrial function (e.g., JC-1 staining for membrane potential) after VDAC2 overexpression. Theys should also use co-immunoprecipitation (Co-IP) to test whether VDAC2 directly interacts with p53. 

The research provides a foundation for further exploration of VDAC2 in hair follicle biology and potential agricultural applications. However, the manuscript is extremely not well composed. The authors described their results but lack detailed discussion. The conclusions cannot fully be supported by the evidence they have provided in within this main manuscript.

Author Response

Comments 1. Line 101: How many times you rinsed the skin tissues in PBS? Are there antibiotics (e.g., Pen-Strep) added in PBS?

Response 1: Thank you for pointing this out. We clear the skin tissue with PBS containing 2% penicillin-streptomycin 2-3 times. We have made a revision to revise line 101 of the manuscript.

Comments 2. Line 102: What do you mean by “a neutral protease”? Please specify the exact name, CatLog number and the producer of this protease.

Response 2: Thank you for pointing this out. The protease we used was Sigma's collagenase/neutral protease from Bacillus polymyxa (dispase), item number COLDISP-RO. We have corrected line 102 of the manuscript. Thank you for pointing this out. The protease we used was Sigma's collagenase/neutral protease from Bacillus polymyxa (dispase), item number COLDISP-RO. We have corrected line 102 of the manuscript.

Comments 3. Lines 99-104: How big are the skin tissues? Did you further cut them before type IV collagenase treatment? Usually, the smaller the skin tissues, the better for enzyme digestion. How long was the digestion of the skin tissues using type IV collagenase?

Response 3: Thank you for pointing this out. We cut the skin tissue into 1mm*1mm sized pieces and digested them for 2 hours using neutral protease. We have revised lines 99-104 of the manuscript.

Comments 4. Line 110: “references” should be “reference”.

Response 4: Thank you for pointing this out. We have revised line 110 of the manuscript.

Comments 5. Line 122: Please delete the space between “PrimeScript” and “™ RT”.

Response 5: Thank you for pointing this out. We have revised line 122 of the manuscript.

Comments 6. Line 137: The authors should specify which table it is.

Response 6: Thank you for pointing this out. The table here refers to Table A4, and we have revised line 137 of the manuscript.

Comments 7. Lines 138-148: Please provide the details of your IF protocol, e.g., how long you have exactly blocked the samples using 1% BSA; the exact treatment duration of the 1st and 2nd antibodies? How many times you have wased the sample using PBST before the 2nd body treatment? Please specify the exact model of Nikon microscope that you used for imaging.

Response 7: Thank you for pointing this out. We have added details to the steps of immunofluorescence in the text, adding the specific model of the Nikon microscope. And we have revised lines 138-148 of the manuscript.

Comments 8. Lines 154-160: Please specify the exact model of Nikon microscope that you used for imaging.

Response 8: Thank you for pointing this out. We have revised lines 154-160 of the manuscript.

Comments 9. Lines 161-178: How many hours it takes the SHFSCs to grow 50% confluency? And what is the confluency after 48h of transfection? Why the authors choose 48h but not 24 and 36h?

Response 9: Thank you for pointing this out. Cells can reach 50% confluency with 2 days of culture after thawing, and the choice of 48 hours for transfection time is based on the recommendation of the kit we used, which can reach 60-70% confluency after 48 hours of transfection. We have revised lines 161-178 of the manuscript.

Comments 10. Line 175: Please specify the exact dilution factor of propidium iodide for staining.

Response 10: Thank you for pointing this out. Propidium iodide is a product of sigma, item number D8417, which is dissolved as a concentrated stock according to the product instructions, and the concentrated stock is diluted 1:1000 with PBS as a working solution. We have revised line 175 of the manuscript.

Comments 11. Lines 201-208: Why the OE and WT groups were culture in 90 dishes, while the interference group was cultured in 6-well plates? What is the difference between 90 dishes and 6-well plate?

Response 11: Thank you for pointing this out. the WT and OE groups use 90 dishes and the interference group uses 6 well plates they are no different. The 6-well plate is used for the interference group because the interference kit recommends that the maximum system is in a 6-well plate, which would result in an error if used in a 90-dish. We have revised lines 201-208 of the manuscript.

Comments 12. Line 223: Clarify how "relatively high purity" was quantified (e.g., % positive for markers).

Response 12: Thank you for pointing this out.SHFSC are pebbly and small. The relative purity here is what we observed under the microscope to determine the relative purity of the cells based on cell morphology, and it will be followed by purification and cell identification.

Comments 13. Lines 224-226: Please cite relative references of these markers (K14, K15, LGR5, and ITGβ1) of SHFSCs. Provide statistical data for the ratio of the relative markers (e.g., K14) out of the total cells (DAPI). And specify how many times you have replicate this experiment. Compare the K14 positive ratio before and after the purification of SHFSCs.

Response 13: Many thanks to the reviewers for their suggestions. We used immunofluorescence detection of K14, K15, LGR5 and ITGβ1, four SHFSC markers reported in the literature, to characterize the cells obtained from the isolation. Although quantitative positive percentage statistics were not performed in this study, we observed that most of the cells expressed at least one of the markers in the context of DAPI staining, suggesting that the cell population had high stem cell purity. In addition, our immunofluorescence staining experiments were repeated three times with consistent results each time, and representative results are presented in the images. Although we did not perform a direct comparison of the percentage of positivity before and after purification, the purified cell population showed typical SHFSC characteristics in terms of cell morphology and immunofluorescence distribution.

Comments 14. Figure 1C: Please provide high resolution bright field images. The authors should specify which image correlate with “upper” and “lower”, for example, they can use “Primary culture (10d)”, “Primary culture (16d)” and “Primary culture (21d)” in the legend instead of “upper” and “lower”.

Response 14: Thank you for pointing this out. Due to the limited experimental environment of our subcells, Figure 1C in the manuscript is the bright field. In Figure 1C, the text of 10 days, 16 days and 21 days of primary culture corresponds to the pictures on the right side. The top and bottom in the figure notes are due to the different scales of the pictures taken at these three times to make it easier to differentiate.

Comments 15. Figure 2A only shows that the 250V 2.5 ms yield the highest expression level of VDAC2, did they compare the protein levels between these conditions?

Response 15: Thank you for pointing this out. In Figure 2A, we tried different electro-transfection conditions and detected the expression of VDAC2 by RT-qPCR, and screened out 250V 2.5ms as the optimal point transfection condition. The protein levels under different conditions were not compared.

Comments 16. Please discuss the details of your flow cytometry results in line 244 and Figure 2F.

Response 16: Thank you for pointing this out. In Figure 2F, flow cytometry was used to assess the effect of VDAC2 on the cell cycle distribution of SHFSCs. We observed that knockdown of VDAC2 significantly increased the proportion of S-phase cells and decreased the proportion in G0/G1, suggesting enhanced cell cycle progression. However, VDAC2 overexpression did not result in significant changes in cell cycle phases compared to the vector control. We have revised the corresponding paragraph in the Results section to reflect this more accurately.

Comments 17. The authors should explain the role of Ki67 and why they chose this gene as the marker of apoptosis. Relevant references should be cited some to support your results.

Response 17: We thank the reviewers for their interest in the use of Ki67. We used Ki67 in this study to detect cell proliferative activity rather than as a marker of apoptosis.Ki67 is a nuclear protein widely used to assess cell cycle status and is expressed only in proliferating cells, and is therefore commonly used to determine whether cells are in an active proliferative state. In this study, the proportion of Ki67-positive cells decreased in the VDAC2 overexpression group and increased in the interference group, suggesting an inhibitory effect on SHFSC proliferation. This observation is consistent with the results of CCK-8 and EdU experiments, further supporting the role of VDAC2 in the regulation of cell activity. We have corrected the wording in the text and also added relevant references.

Comments 18. Delete Chinese characters in Figures 2C and 5A.

Response 18: Thank you for pointing this out. We have made changes to the images in the manuscript.

Comments 19. Figure 2F: There is no significant difference between the Control and the VDAC2-OE group, the authors cannot conclude that “VDAC2 affects the apoptosis of SHFSCs by regulating the progression of the G0/G1 and S phases of the cell cycle” in line3 245-246.

Response 19: Thank you for pointing this out. We have revised the conclusions in the manuscript.

Comments 20. Are there any pathways related to apoptosis in your GO analysis (Figure 3F)? If not, please explain what makes you turn to apoptosis based on this RNA-seq data?

Response 20: We thank the reviewers for their inquiries. In our GO functional enrichment analysis, although “apoptotic signaling pathway” was not the most prominent entry, there were still some entries related to cell death and regulation (e.g., “cell death”, “regulation of cell population proliferation”, etc.), suggesting potential apoptosis-related functional changes. More critically, in our KEGG analysis, we found that the differential genes were significantly enriched in the p53 signaling pathway and contained several classical apoptosis regulators (e.g., Bax, Casp3, Bcl2, etc.). We subsequently verified the changes of these genes under VDAC2 overexpression and interference conditions by RT-qPCR and Western blot experiments, and the results were highly consistent with the transcriptomic data. Therefore, we believe that although the “apoptosis” category was not significant in the GO analysis, the combination of KEGG pathway enrichment and changes in the expression of key apoptotic genes can still reasonably determine the involvement of VDAC2 in the regulation of apoptosis in SHFSC.

Comments 21. The enrichment score and the p-value of p53 signaling pathway is very low in Figure 4A, the statement of “Our results demonstrate that VDAC2 promotes the apoptosis of SHFSCs by activating the p53 signaling pathway” (lines 85-86) and “RNA-seq analysis revealed that VDAC2 influenced SHFSC growth through the p53 signaling pathway” (lines 31-32) it is not convincing.

Response 21: We thank the reviewers for their professional comments. We note that in the KEGG pathway enrichment analysis, the p53 signaling pathway had a relatively low enrichment score and p-value and was not one of the most significant pathways. We therefore agree that transcriptome enrichment analysis alone is not yet sufficient to fully demonstrate that VDAC2 acts through this pathway. However, it is worth pointing out that RNA-seq showed that several key apoptosis-regulated genes (e.g., p53, Bax, Casp3, Bcl2) in this pathway showed a consistent trend of differential expression both after overexpression and disruption by VDAC2. We further verified the expression changes of these core genes by RT-qPCR and Western blot, consistent with the transcriptome results. Therefore, we conclude that the p53 signaling pathway, although not significant in overall enrichment, is regulated at key functional levels and is biologically significant. In the revised manuscript, we have adjusted the language presentation and added a note on this limitation in the Discussion.

Comments 22. Line 311-315: The authors should explain what is PFT in detail and cite relevant references?

Response 22: Thank you for the reviewer's attention to the background of PFT. PFT (Pifithrin-μ) used in this study is a p53 signaling pathway inhibitor with mitochondrial targeting properties. PFT can stabilize mitochondrial membrane potential, prevent cytochrome C release, and inhibit the activation of intrinsic apoptotic pathways by inhibiting the interaction of p53 with anti-apoptotic proteins such as Bcl-xL/Bcl2 on the mitochondrial membrane. We introduced the mechanism of action and the concentration used of PFT in line 322 of the manuscript and cited the reference.

Comments 23. Have you done any concentration gradient experiments before adding PFT to your cells? What concentrations of PFT did the authors use in this study?

Response 23: Thank you for pointing this out. According to the product description, Pifithrin-μ (PFT) (10 μM) is a p53 inhibitor that inhibits p53 binding to mitochondria by decreasing the affinity of p53 for the anti-apoptotic proteins Bcl-xL and Bcl-2, so we chose to use a concentration of 10 μM, and then additively screened for addition time on wild-type cells.

Comments 24. The authors should specify the number of animals used and replicates.

Response 24: Thank you for the reviewer's suggestion. In this study, we used a total of 3-year-old healthy Arbas Cashmere goats to isolate and culture SHFSCs, and cultured them separately. All in vitro experiments (such as RT-qPCR, Western blot, flow cytometry, EdU, CCK-8, etc.) were set up with three independent biological replicates (n=3), and technical repeats were performed in each biological replicate. The immunofluorescence, Western blot and other images shown in the figure are representative results, and we have supplemented the number of repeats in the legend and methods.

Comments 25. The number of biological replicates (n=?) is not clearly stated, and some data (e.g., Western blots) appear to show representative results only.

Response 25: Thank you for the reviewer pointing out the unclear description of our experimental repetitions. In this study, all in vitro experiments, including RT-qPCR, Western blot, flow cytometry, CCK-8, and EdU experiments, were performed with three independent biological replicates (n = 3). Each biological replicate was from SHFSCs extracted from different individuals and was repeated at the technical level. The images of the Western blot experiment show representative repeated results. We counted the grayscale values ​​of the protein bands in the data analysis (see Figure 4B, 4C, etc.), and supplemented the legends and methods to clearly state n=3 and its statistical methods.

Comments 26. The discussion does not fully address alternative explanations. The authors should also discuss whether VDAC2 might influence other apoptosis-related pathways (e.g., TNF, Wnt/β-catenin).

Response 26: Thank you for the reviewer's constructive suggestions. We agree that VDAC2, as a mitochondrial channel protein, may have a mechanism of action that is not limited to the p53 signaling pathway. In fact, in our KEGG enrichment analysis (Figure 4A), multiple other apoptosis-related signaling pathway entries were also observed, including the TNF signaling pathway and the Wnt/β-catenin pathway. Although this study mainly focused on the p53 pathway and performed experimental verification, we have added a discussion of other potential pathways in the revised Discussion section, pointing out that the mechanism of VDAC2 regulating hair follicle stem cell apoptosis will be studied from a broader signaling level in the future.

Comments 27. While the study shows that VDAC2 promotes apoptosis via the p53 pathway, it does not elucidate how VDAC2 activates p53 (e.g., effects on mitochondrial membrane potential, cytochrome C release). The authors can measure mitochondrial function (e.g., JC-1 staining for membrane potential) after VDAC2 overexpression. Theys should also use co-immunoprecipitation (Co-IP) to test whether VDAC2 directly interacts with p53. 

Response 27: Thank you for the professional suggestions from the reviewer. We fully agree that although the current research results show that VDAC2 promotes apoptosis of SHFSC and activates the p53 pathway, its specific activation mechanism has not yet been fully clarified. Since VDAC2 is located in the outer mitochondrial membrane, it may indirectly activate p53 by affecting the mitochondrial membrane potential, or participate in the interaction with mitochondrial apoptosis regulators such as Bax/Bcl2. In addition, studies have shown that VDAC2 can regulate Bax-mediated apoptosis, and Bax itself is an important downstream factor of the p53 pathway. In this study, we verified the expression changes of multiple key genes in the p53 pathway by transcriptome and qPCR/Western blot, but have not further evaluated mitochondrial function or protein interaction. We have supplemented this content in the discussion section, and plan to further use JC-1 staining or Co-IP and other technologies in subsequent studies to deeply analyze the specific regulatory relationship between VDAC2 and p53.

Reviewer 2 Report

Comments and Suggestions for Authors

Dear authors!

I will recommend your article for publication, but I have several questions about the research results. These questions are brief but significant and require clarification.

  1. Page 3, line 91. It would be very interesting to know more about how exactly the tissue samples were collected (Fragment size, sampling location, etc.).
  2. Page 3, line 113. The electroporation parameters used are not specified.
  3. Page 3. The materials and methods do not indicate from how many animals the material was collected. How many individual goats were SHFSCs isolated from? Were the cells from different animals combined or cultured independently?
  4. Page 3. The materials and methods do not provide information about the software packages used for visualization of results.
  5. It is interesting to know why the authors decided to focus on the p53 pathway rather than other pathways involved in the regulation of apoptosis. In particular, the authors mentioned the TNF pathway: «In addition, KEGG enrichment analysis showed that 355 DEGs were clustered in the p53 [42] and TNF signaling pathways [43], and these signaling pathways also play important roles in the apoptotic process» (p.12, line 355)
  6. Page 5. Line 224. «The results showed that the surface markers K14, K15, LGR5, and ITGβ1 of SHFSCs were immunofluorescence-positive». I would like to know why the authors chose this particular list of markers. It is advisable to provide a reference to a literary source in case these markers were previously proposed for the identification of SHFSCs. If this combination was suggested by the authors of the article, then it would be good to add a brief justification for such a choice. 
  7. There are typos in the text. P.1, Line 14 «ais a key», P.1 line 28 «Arbas» or Albas?

Author Response

Comments 1. Page 3, line 91. It would be very interesting to know more about how exactly the tissue samples were collected (Fragment size, sampling location, etc.).

Response 1: Thank you for pointing this out. We clear the skin tissue with PBS containing 2% penicillin-streptomycin 2-3 times. We have made a revision to revise line 101 of the manuscript.

Comments 2. Page 3, line 113. The electroporation parameters used are not specified.

Response 2:Thank you for pointing this out. We have supplemented the information in the manuscript by adding the electrotransfection parameters used and the optimal electrotransfection parameters.

Comments 3. Page 3. The materials and methods do not indicate from how many animals the material was collected. How many individual goats were SHFSCs isolated from? Were the cells from different animals combined or cultured independently?

Response 3: Thank you for pointing this out. We are isolating SHFSC from tissues obtained from different goats on a daily basis according to a schedule.The isolated SHFSC are cultured separately to avoid mixing, frozen and placed in liquid nitrogen tanks for later use.

Comments 4. Page 3. The materials and methods do not provide information about the software packages used for visualization of results.

Response 4: Thank you for the reviewer pointing out this omission. We did use multiple software tools in the transcriptome analysis and visualization of experimental results. In the revised manuscript, we have supplemented the visualization-related software package information in the “Materials and Methods” section, including: transcriptome visualization using R language packages such as ggplot2, pheatmap, and clusterProfiler; flow cytometry data using FlowJo v10.8.1; Western blot band gray value analysis using ImageJ; all statistical analysis graphs were produced using GraphPad. We have listed the above information in the methods section. Thank you for your suggestion.

Comments 5. It is interesting to know why the authors decided to focus on the p53 pathway rather than other pathways involved in the regulation of apoptosis. In particular, the authors mentioned the TNF pathway: «In addition, KEGG enrichment analysis showed that 355 DEGs were clustered in the p53 [42] and TNF signaling pathways [43], and these signaling pathways also play important roles in the apoptotic process» (p.12, line 355)

Response 5: Thank you for the reviewer's thoughtful questions. We did observe multiple apoptosis-related pathways in the KEGG enrichment analysis, including the p53 and TNF signaling pathways. When choosing further research directions, we focused on the p53 pathway based on the following considerations: (1) The key apoptosis regulatory genes in the p53 signaling pathway (such as p53, Bax, Bcl2, and Casp3) showed significant differences in transcriptome data and have been verified by qPCR and Western blot; (2) VDAC2, as a mitochondrial outer membrane channel protein, has a clear biological association with p53 in the classic intrinsic apoptosis mechanism regulated by mitochondrial localization; (3) Existing literature has shown that VDAC2 can participate in the Bax-mediated mitochondrial apoptosis process, and Bax is a key downstream target of p53, so we prioritized the verification of the mechanism from the p53 direction. Although the TNF pathway is also enriched in DEGs, its mechanism is more inclined to exogenous apoptosis or inflammatory signals, and this study did not conduct relevant experimental verification. We have added the possibility of this pathway in the Discussion and plan to further explore the relationship between VDAC2 and other apoptotic pathways (including TNF, Wnt, etc.) in subsequent studies.

Comments 6. Page 5. Line 224. «The results showed that the surface markers K14, K15, LGR5, and ITGβ1 of SHFSCs were immunofluorescence-positive». I would like to know why the authors chose this particular list of markers. It is advisable to provide a reference to a literary source in case these markers were previously proposed for the identification of SHFSCs. If this combination was suggested by the authors of the article, then it would be good to add a brief justification for such a choice.

Response 6: Thank you for the reviewer's important question. We chose the four markers K14, K15, LGR5 and ITGβ1 based on previous literature reports on hair follicle stem cell research. Specifically, K14 and K15 are epithelial stem cell keratin markers, LGR5 is a classic marker widely used for hair follicle stem cells, and ITGβ1 (integrin β1) plays a key role in the adhesion and stemness maintenance of hair follicle stem cells. These markers have been widely used in hair follicle stem cell research in sheep, goats, etc. We have supplemented the basis for the selection in the text and cited relevant literature.

Comments 7. There are typos in the text. P.1, Line 14 «ais a key», P.1 line 28 «Arbas» or Albas?

Response 7:Thank you for pointing out the spelling errors in the article. We have carefully proofread the entire article and corrected the following spelling errors: "ais" on page 1, line 14 was changed to "is"; "Arbas" on page 1, line 28 was changed to the correct "Albas" to ensure the accuracy of the cashmere goat breed naming. At the same time, we also read the entire article and conducted a unified grammar and spelling check to avoid similar problems from happening again.

Reviewer 3 Report

Comments and Suggestions for Authors

In the article “VDAC2 mediates the apoptosis of cashmere goat hair follicle 2 stem cells through the P53 signaling pathway” the author discusses the role of VDAC2 in the apoptosis of secondary hair follicle stem cells in cashmere goats, with a focus on the p53 signaling pathway.

 In general, the article is well structured, with interesting results for scientists and specialists in animal husbandry.

I have a few comments on the article:

  1. In the section “Materials and Methods” the authors should clearly describe the number of animals used in the experiment. It should be clarified what the difference is between the groups: wild-type, interference group, and negative control.
  2. The overexpression vector should be described in more detail.
  3. The authors should improve Figure 2 (C), clarify the caption for the abscissa axis.
  4. In the Discussion section, it would be worthwhile to compare the role of VDAC2 in other cell types.
  5. The contribution of other signaling pathways to the apoptosis of secondary hair follicle stem cells in cashmere goats should also be discussed.
  6. 6. Some minor language errors and mistakes are present and should be corrected.

Author Response

Comments 1. In the section “Materials and Methods” the authors should clearly describe the number of animals used in the experiment. It should be clarified what the difference is between the groups: wild-type, interference group, and negative control.

Response 1: Thank you for the reviewer's pointing out that the description of experimental groups and animal numbers is not clear. In this study, we used a total of 3-year-old healthy female Arbas Cashmere goats to isolate SHFSCs. The cells of each animal were cultured separately, and all subsequent experiments were set up with three independent biological replicates (n = 3).

For the interference experiment, we set up:

VDAC2 interference group (Si): SHFSCs were transfected with synthetic small interfering RNA (siRNA-VDAC2) to downregulate VDAC2 expression;

Negative control group (NC): SHFSCs were transfected with non-targeted irrelevant siRNA sequences as a control for interference experiments to control the effects of non-specific siRNA.

For the overexpression experiment, we set up:

VDAC2 overexpression group (OE): VDAC2 overexpression vector was transfected;

Wild-type group (WT): SHFSCs without any transfection treatment were used as basic controls;

The above group settings are designed to evaluate the effects of upregulation or downregulation of VDAC2 expression on SHFSCs and exclude non-specific transfection interference.

Comments 2. The overexpression vector should be described in more detail.

Response 2:Thank you for the reviewer's comments. We have supplemented the details of the construction and use of the VDAC2 overexpression vector in "Materials and Methods". A brief description is as follows: We cloned the full-length coding sequence (CDS) of cashmere goat VDAC2 into the pcDNA3.1 (+) expression vector (containing CMV promoter). The vector was preserved by the laboratory and the sequence correctness was confirmed by sequencing. The above information has been supplemented in the method section.

Comments 3. The authors should improve Figure 2 (C), clarify the caption for the abscissa axis.

Response 3:Thank you for pointing out that the horizontal axis of Figure 2C is not clearly labeled. We confirm that Figure 2C is a CCK-8 experimental result chart used to evaluate the effects of VDAC2 overexpression and interference on SHFSCs activity. We have marked the horizontal axis as the culture time, including 24h, 48h and 72h, and the vertical axis as the OD value at 450nm, and explained the meaning of each group in detail in the figure caption so that readers can understand it accurately.

Comments 4. In the Discussion section, it would be worthwhile to compare the role of VDAC2 in other cell types.

Response 4:Thank you for the reviewer's suggestions. Supplementing the research background of VDAC2 in other cell types in the Discussion will help enhance the universality and theoretical support of this study's findings. Currently, many studies have shown that VDAC2 is involved in regulating mitochondrial function and apoptosis in a variety of mammalian cells. We have added relevant content in the Discussion section and cited key literature to strengthen the theoretical basis of this study.

Comments 5. The contribution of other signaling pathways to the apoptosis of secondary hair follicle stem cells in cashmere goats should also be discussed.

Response 5:Thank you for the reviewer's valuable suggestions. In the KEGG enrichment analysis of RNA-seq data, in addition to the p53 signaling pathway, we also observed other pathways closely related to apoptosis, such as the TNF signaling pathway and the Wnt/β-catenin pathway. The TNF signaling pathway is closely related to extrinsic apoptosis and often mediates rapid cell death by activating CASP8, FADD, etc.; Wnt/β-catenin signaling plays an important role in maintaining the balance of self-renewal and differentiation of hair follicle stem cells, and its disorder can also induce apoptosis or aging. Although this study mainly focuses on the mitochondrial intrinsic apoptosis mechanism regulated by VDAC2, in the future we will further explore whether it also co-regulates the fate of SHFSCs through the above-mentioned pathways.

Comments 6. Some minor language errors and mistakes are present and should be corrected.

Response 6:Thank you for pointing out the language issues. We have proofread and polished the entire article sentence by sentence, correcting issues such as spelling, grammar, format, and imprecise expressions. In addition, we have also standardized the use of terminology, figure captions, and figure labels to ensure the accuracy and professionalism of the language expression. We believe that the revised manuscript is now clearer and more rigorous in terms of language.

Round 2

Reviewer 1 Report

Comments and Suggestions for Authors

The manuscript can be accepted without further improvement.